# Evaluation of biochemical, physiological traits and percentage of essential oil of sickleweed (*Falcaria vulgaris*) population in different geographical and climatic regions

**Mehdi Rahimi**[1]*, **Mojtaba Kordrostami**[2]*, **Jaber Nasiri**[2]

**1** Department of Biotechnology, Institute of Science and High Technology and Environmental Sciences, Graduate University of Advanced Technology, Kerman, Iran, **2** Nuclear Science and Technology Research Institute (NSTRI), Nuclear Agriculture Research School, Karaj, Iran

* me.rahimi@kgut.ac.ir, mehdi83ra@yahoo.com (MR); mkordrostami@aeoi.org.ir (MK)

**Data Availability Statement:** All relevant data are within the paper.

## Abstract

Sickleweed (*Falcaria vulgaris*) is the name of a species of annual, and perennial herbaceous plants of the genus *Falcaria*. Climate change could negatively influnces the performance of various plant species in plant kingdom. In this study, 15 different sickleweed populations from seven provinces of the country were collected based on an unbalanced nest design with 10 replications and the percentage of essential oil, types of chlorophyll, phenol, proline, protein, and carotenoids were measured on them. The results showed that there was a significant difference between populations at the level of one percent for the studied traits. The results of the mean comparison showed that populations Ard-Shaban and Qaz-Ilan in terms of the evaluated traits and especially the percentage of essential oil were at the upper of the studied samples and selected as suitable populations. In addition, populations Gilan-Deylaman and Kur-Gerger-e Sofla were also identified as superior populations in terms of studied traits by cluster analysis and principle component analysis (PCA). Since the high level of proline and biochemical and physiological traits in plants can play a role in plants' tolerance to stresses, therefore, populations with high values of these traits can be used in stress tolerance breeding programs. Therefore, in this study, populations Gilan-Deylaman and Kur-Gerger-e Sofla can be suitable populations for this purpose. In addition, the essential oil of this plant is used in the treatment of diseases, and therefore populations Ard-Shaban and Qaz-Ilan, which showed a high percentage of essential oil, can be used in breeding programs to increase the performance of this trait.

## Introduction

Today, the nutritional, health, and economic value of medicinal plants, along with the role of 25% in the production of drugs in the world, has led to considerable attention to them [1]. The country of Iran has a variety of plant species due to its climatic diversity. The genetic resources

**Funding:** This work was supported by the Iran National Science Foundation (INSF) under grant number [99026928]. The funders had no role in study design, data collection and analysis, decision to publish, or preparation of the manuscript.

available in each country are one of the most valuable potential agricultural resources of that country. The main goal of studying the diversity between and within different plant populations and the relationship between germplasm collections is to finally use this information for the production and development of cultivars with better productivity than cultivated species [2–4]. In general, genetics affects plant yield and secondary metabolites of medicinal plants, and the interaction of area×mass is very significant on the yield of total dry matter and plant and varies depending on the region and type of mass [5–7]. Transferring masses and different species from one region to another as well as the forcing of the adaptation plant to the new environment can change the ratio of secondary metabolites and in some cases induce new medicinal metabolites [8].

Climate, directly and indirectly, affects all factors affecting the environment and plant growth [9,10]. Experiments conducted in recent years show the effective role of climatic factors on the distribution of plants [11]. According to some studies, although the weather factor plays the most important role in the growth and distribution of plants, plants are developed through several different factors including climate, soil characteristics, and natural conditions [12]. Sickleweed (*Falcaria vulgaris*) belongs to the *Falcaria* genus and the Apiaceae family and belongs to the Eudicots Clade. Although Sickleweed has been described as annual, biennial or perennial [13], but according to the authors' observations and evaluations, it was annual. Sickleweed is especially important because of its nutritional value and medicinal effects on the human body. This plant is usually consumed as a vegetable and is a common food in spring. It is recommended in traditional medicine books to treat skin wounds, stomach problems, liver diseases, and blood purification, and its therapeutic effects have been proven in modern medical science [13]. The increasing recognition of the properties of this plant species and the increasing population growth along with the decrease in rainfall have put increasing pressure on this plant population that preventive actions should be done for the protection and breeding of this plant.

There are different methods to investigate the diversity between different plant samples; one of the cheapest and most common methods is the morphological diversity investigation. Although morphological diversity is influenced by environmental conditions compared to molecular diversity, it is widely used in different plants. Identification of morphological diversity is not only useful in the management of plant germplasms but also provides a good idea to researchers for plant breeding [14]. Moreover, one of the important goals of medicinal plants is to select populations with high levels of secondary metabolites and increase the number of important compounds in these plants, which is possible by identifying effective physiological and biochemical parameters and knowing the relationships between plants and environmental factors [15]. Geographical and climatic factors are effective in the production of secondary metabolites and morphological characteristics of medicinal plants. Although effective substances are made by guiding genetic processes, their production is significantly influenced by environmental factors such as light, temperature, and altitude. Environmental factors affect the total amount of effective substances, their constituent elements, dry weight production, and plant morphology [16].

Various studies have been conducted in the field of investigating biochemical and physiological traits under the influence of various factors such as different regions, different environmental conditions, etc., in different medicinal plants [17–20]. In the case of the medicinal species Satureja, Spear Thistle, Persian hogweed, and chicory, it was shown that there is a direct relationship between the increase in height and the amount of phenolic and flavonoid active substances [21]. In the study of the genetic diversity of the this population with SCoT molecular marker and morphological traits, the population was divided into three and two groups based on the markers and traits, respectivlley [22].

Research have revealed that there might be significant regional variation in the biochemical characteristics of different plants. For instance, it has been discovered that arid regions with little rainfall and high temperatures have higher levels of medicinal plants in terms of their total phenolic and flavonoid content, antioxidant activity, and essential oil content. The influence of environmental elements like temperature, rainfall, and soil type can be used to explain the diversity in the biochemical characteristics of medicinal plants across various geographical areas [23–27].

So far, not much study has been done in the world and Iran regarding the evaluation of morphological traits of different populations of sickleweed plants and the determination of their morphological identification. The biochemical and physiological traits of sickleweed vary across different populations, depending on the environmental factors such as temperature, rainfall, soil type, and altitude. Therefore, this study aims to evaluate the effect of geographical origin on populations and to investigate the diversity between populations, as well as to identify the best populations with breeding potential with the highest yield in terms of important traits such as essential oil, and physiological and biochemical traits. In addition, the use of this information is for creating a future protection program to prevent them from becoming extinct.

## Materials and methods

### Climatic factors

The climatic characteristics of the regions are shown in Table 1. The information related to the height of the location as well as the geographic longitude and latitude was recorded by Google Earth software, and the information related to the weather as well as various bioclimatic parameters [28] was estimated from the WorldClim website.

### Plant collection

The 15 different plant samples (Table 1) were collected from seven provinces of Iran. These seven provinces include Ardabil, Kurdistan, Kermanshah, Gilan, Hamedan, Qazvin, and Qom so the number of selected regions was different for each province. For each population in each

**Table 1. Characteristics of the collection areas of Sickleweed plant samples.**

| Row | Province | City | Region | Longitude (degree) | Latitude (degree) | Height above sea level (meter) |
|---|---|---|---|---|---|---|
| G1 | Kermanshah | Sonqor | Kivananat | 47.27889 | 34.84972 | 1847 |
| G2 | Kermanshah | Sonqor | Bavaleh | 47.69778 | 35.02722 | 2010 |
| G3 | Kermanshah | Sahneh | Sahneh | 47.75833 | 34.44194 | 1465 |
| G4 | Hamedan | Asad abad | Chaharduli | 48.065 | 34.93139 | 1890 |
| G5 | Kurdistan | Gorveh | Panjeh Ali | 47.70834 | 35.18667 | 1912 |
| G6 | Kurdistan | Dehgolan | Bolbanabad | 47.41195 | 35.15222 | 1851 |
| G7 | Kurdistan | Dehgolan | Amirabad | 47.31084 | 35.10361 | 1974 |
| G8 | Kurdistan | Kamyaran | Gerger-e Sofla | 47.27667 | 35.00334 | 1824 |
| G9 | Kurdistan | Kamyaran | Qaleh Gah | 47.21196 | 34.95001 | 1611 |
| G10 | Kurdistan | Bijar | Nemat abad Auliya | 47.41333 | 35.68444 | 1922 |
| G11 | Kurdistan | Bijar | Seylatan | 47.83054 | 36.03666 | 1617 |
| G12 | Qazvin | Qazvin | Ilan | 50.64778 | 36.42528 | 1450 |
| G13 | Ardabil | Meshkinshahr | Shaban | 47.44833 | 38.37556 | 1236 |
| G14 | Qom | Qom | Khalajastan | 50.19028 | 34.67944 | 1952 |
| G15 | Gilan | Siahkal | Deylaman | 49.90528 | 36.88889 | 1455 |

region, 10 different samples were selected and considered as replicates. In addition, the evaluation of traits and measurements was performed on these samples, and studies were conducted based on a nesting design with an unbalanced number of regions. In each area, the distance of the samples from each other was at least 10 meters to avoid interference from the samples.

## Sample preparation

The leaves of the plants were collected in the vegetative stage of the Sickleweed plant in the early and mid-spring of 2021, and the fresh leaves were placed in foil after harvesting, placed in liquid nitrogen, and transferred to the laboratory for use in the next experiments such as percentage of essential oil, measuring physiological and biochemical traits based on methods described by Sudhakar et. al, [29].

## Chlorophyll content

Total chlorophyll, as well as chlorophylls a and b, were measured by Arnon's method [30] using 0.2 g of fresh plant leaf material in a test tube and grind with 10 mL of 80% acetone, and the absorbances read at 663 and 645 nm in a spectrophotometer. Calculate the amount of chlorophyll present in the extract in mg chlorophyll per g tissue using the following equations as described by Sudhakar et. al, [29]:

$$mg\ chlorophyll\ a/g\ tissue = [(12.7 \times A663) - (2.69 \times A645)] \times \frac{v}{1000 \times w} \qquad (1)$$

$$mg\ chlorophyll\ b/g\ tissue = [(22.9 \times A645) - (4.68 \times A663)] \times \frac{v}{1000 \times w} \qquad (2)$$

$$mg\ total\ chlorophyll/g\ tissue = [(20.2 \times A645) - (8.02 \times A663)] \times \frac{v}{1000 \times w} \qquad (3)$$

Where:
A = absorbance at specific wavelengths
V = final volume of chlorophyll extract
W = fresh weight of tissue extracted

## Total carotenoids

The total carotenoids were measured by Price and Hendry's method [31] using the sample extract, obtained for the estimation of chlorophylls by the Acetone Method. The absorbances are read at 663 nm, 645 nm, and also at 480 nm in a spectrophotometer and the concentration of pigments is mg/g fresh weight. Carotenoid content is calculated using the following equations as described by Sudhakar et. al, [29]:

$$mg\ total\ carotenoids/g\ tissue = [A480 + (0.114 \times A663) - (0.638 - A645)] \times \frac{v}{1000} \times w \ (4)$$

This equation compensates for the interference at this wavelength, due to chlorophyll.

## Proline

The Bates et al. method [32] was used to measure the proline content of leaves. Initially, 2 ml of the supernatant after centrifuging the extract was combined with 2 ml of the reagent ninhydrin and 2 ml of pure acetic acid, then the mixture was placed in an air bath at 100°C for 1 h. The tubes containing the mixture were then as soon as possible put in the ice bath. The

mixture was then given 4 ml of toluene, and the tubes were thoroughly vortexed. The tubes were stacked for 15 to 20 min to create two distinct layers. Proline concentration was determined using the toluene and proline-containing upper color phase. The absorbance was determined at 518 nm and the amount of proline was calculated using a standard curve.

## Protein

The proteins were extracted from the aerial parts at a temperature between 0°C and 4°C in ice water with phosphate buffer. Then, the Coomassie Brilliant Blue (CBB) G-250 was used to give a consistent blue color. After 25 min, the absorbance was read by a spectrophotometer at 595 nm. Protein concentrations were calculated using a standard curve to measure the amount of total protein by Bradford's method [33].

## Reducing sugars

In a Chinese mortar, 0.02 g of the plant's aerial section was ground with 10 ml of distilled water and heated on an electric stove. Once the solution had boiled, the heat was turned off, and filter paper was used to collect the solution. Test tubes were filled with 2 ml of the produced extract, and two milliliters of copper sulfate solution, and then the test tubes' ends were sealed with cotton before being submerged in a hot bath at a temperature of 100°C for 20 min. the 2 ml of the phosphomolybdic acid solution were added to the tubes once they had cooled, and after a short while, a blue tint emerged. The test tubes were then agitated ferociously with a vortex device to evenly spread the color throughout the test tube. Using a spectrophotometer and the standard curve, the absorbance intensity of the solutions at 600 nm wavelength was measured by the Somogyi method [34] to show the amount of reducing sugars.

## Total phenolic

The extraction of phenolic compounds was done according to Malik and Singh [35] and the stages of preparation, extraction, etc. are shown in Sudhakar et. al, [29]. The total amount of phenolic compounds were measured by the Folin-Ciocalteu method at 650 nm spectrophotometrically and expressed as mg phenols/100 g material.

## Essential oil

The percentage of essential oil was also determined by distillation with water using Clevenger-type apparatus by the European Pharmacopoeia [36], and after drying the water with the essential oil using dry sodium sulfate, the amount of essential oil was purified and then the percentage of essential oils was calculated relative to the dry weight.

## Statistical analysis

Statistical analyzes including descriptive statistics, analysis of variance, and multivariate analysis were performed on the data. Descriptive statistics including mean, range, and standard deviation, etc. were performed on the average data of 15 populations, and the phenotypic coefficient variation (PCV) was calculated as follows for the studied traits.

$$\mathrm{CV_p} = \left( \sqrt{\sigma_p^2} / \bar{\mathrm{x}} \right) \times 100 \qquad (5)$$

Before analysis, the variance of the data has checked for the correctness of the assumptions of the variance analysis, such as normal distribution of experimental errors, independence of experimental errors, uniformity of variances within the treatment, and the absence of

interaction between treatment and provinces. After establishing these assumptions, the variance analysis of the data was carried out based on the unbalanced nested design, and Tukey's test was used to compare the mean. In addition, the variance components were calculated based on the expected value of the mean square of the unbalanced nested design. The measured data for the essential oil percentage, protein, sugar content, and proline traits were very small, so the first data were multiplied by 100 and then the obtained data were used and the variance analysis was performed using SAS software version 9.4 [37].

Multivariate analyses were used to investigate the genetic structure and grouping of populations and to identify kinship relationships between them. Grouping of sickleweed populations by using traits as well as grouping of the climatic regions by using bioclimatic parameters were done based on cluster analysis. The cluster analysis was done based on different methods and different distance criteria. Then, the distance method and the criterion that had the highest cophenetic correlation coefficient were selected and the grouping was done based on it. Finally, the distribution of the populations was also done by principal component analysis and the biplot method. The cluster analysis method and principal component analysis were done with PAST software [38]. Determining the number of groups was done with the Factoextra [39] package and R software.

## Results and discussion

Nineteen bioclimatic indicators were measured and calculated for the collection locations in this study to investigate the geographic variety. The diversity and similarity of these areas were then assessed using various cluster analysis techniques and distance criteria. Fig 1 displays the outcomes of the cluster analysis for these areas. The findings demonstrated that these locations

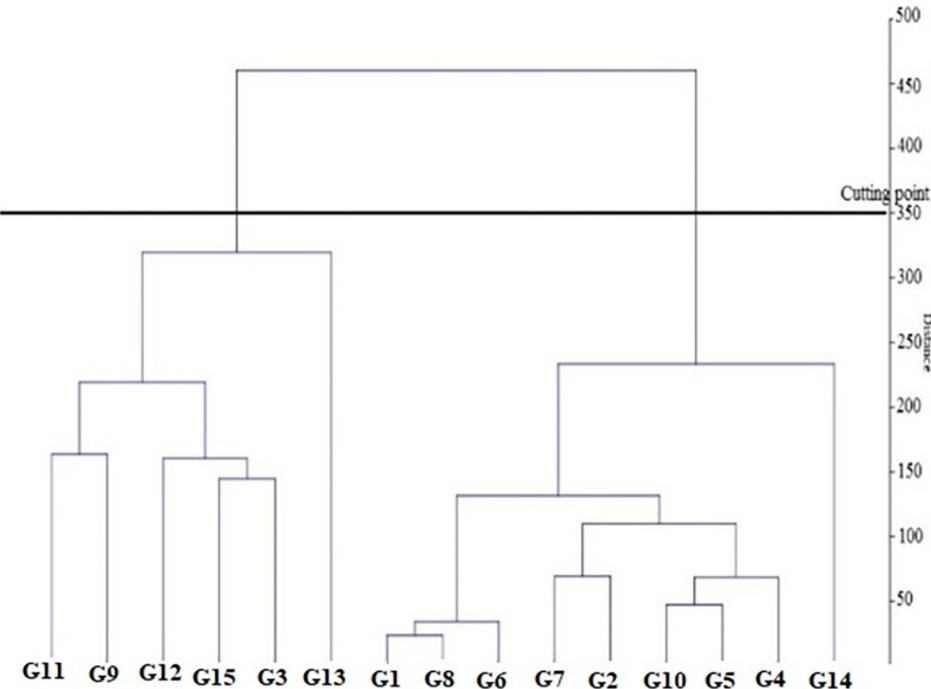

**Fig 1. The region cluster analysis of sickleweed population by bioclimatic parameters.** The name of population is given in Table 1.

are divided into two groups using the Elbow technique and that they share similar weather patterns and bioclimatic data. The Elbow method uses a graphical representation to find the optimal number of clusters by Within-Cluster Sum of Square. The inertia is the sum of the squared distances from each point to the cluster center [40]. In the first group were Seylatan and Qaleh Gah regions from Kurdistan province, Sahneh from Kermanshah, Deylaman from Gilan, Ilan from Qazvin, and Shaban from Ardabil, and in the second group were Kiyunanat and Bavaleh regions from Kermanshah, Panjeh Ali, Bolbanabad, Amirabad, Gerger-e Sofla and Nemat abad auliya from Kurdistan, Khalajastan from Qom and Chaharduli from Hamedan. The range of a species geographic distribution is a complex issue in population ecology that cannot be fully understood. Generally speaking, whether a species can survive in its native environment depends on how well an individual can meet biological needs such as dietary, physiological, and behavioral needs [41]. Weather patterns and terrain, or more specifically, geographic diversity, have an impact on the quantity and richness of resources. The classic hypothesis of allopatric speciation postulates that physical obstacles like distances, lowlands, and heights, among others, can separate populations and lead to various trends in the variety of those populations [42,43]. The alterations that distinct populations have endured along distinct evolutionary pathways may have resulted in a sort of superiority in reproduction and adaption to various places [44]. Plants grown in arid regions with low rainfall and high temperatures may have higher percentages of essential oil due to the need to conserve water and prevent water loss through transpiration [23,27,45,46].

Because the purpose of studying genetic populations is to compare genotypes in terms of the amount of existing variation, This way, a range of data can be used for the initial comparison between the studied genotypes and get a general view of the amount of existing variation. Descriptive statistics including the minimum, maximum, and range of the studied traits are shown in Table 2. The coefficient of variation parameter is one of the most important and valuable indices for estimating diversity in populations, and since this criterion is not affected by the trait measurement unit or its range, it is more important than other diversity criteria such as the range. Therefore, it is possible to make reliable selections to improve traits that have a higher coefficient of variation. Examining the phenotypic coefficient of variation of the traits showed that phenol, Chl. b and essential oil had the highest PCV. Therefore, these traits can be used in the breeding programs and effective selections can be made among the studied sickleweed populations to improve and breed these traits.

In addition, the lowest PCV is related to protein traits, and the improvement of this trait will be less successful than other traits through selection in the studied population. According

**Table 2. Descriptive statistics of biochemical and physiological traits in sickleweed populations.**

| Traits | Statistical parameters | | | | | |
|---|---|---|---|---|---|---|
| | **Min** | **Max** | **Range** | **Avereage** | **Stand. dev** | **PCV** |
| Essential oil (%) | 0.076 | 0.089 | 0.013 | 0.081 | 0.004 | 5.362 |
| Protein (mg/GFW) | 0.00679 | 0.00685 | 0.00006 | 0.00682 | 0.00002 | 0.23020 |
| Proline (mg/GFW) | 0.00245 | 0.00288 | 0.00043 | 0.00273 | 0.00011 | 4.19188 |
| Sugars (mg/GFW) | 0.04699 | 0.05737 | 0.01038 | 0.05530 | 0.00267 | 4.82979 |
| Chl. a (mg/GFW) | 7.381 | 8.397 | 1.017 | 8.014 | 0.364 | 4.545 |
| Chl. b (mg/GFW) | 2.018 | 2.516 | 0.498 | 2.198 | 0.119 | 5.426 |
| Total Chl. (mg/GFW) | 9.597 | 10.893 | 1.296 | 10.235 | 0.397 | 3.884 |
| Carotenoids (mg/GFW) | 4.152 | 4.608 | 0.456 | 4.434 | 0.163 | 3.685 |
| Phenol (mg/100 GFW) | 52.031 | 88.149 | 36.118 | 55.315 | 9.100 | 16.451 |

**Table 3. Unbalanced nest variance analysis of physiological and biochemical traits in sickleweed populations.**

| S.O.V. | Df. | The mean square of traits | | | | | | | | |
|---|---|---|---|---|---|---|---|---|---|---|
| | | Essential oil (%) | Phenol (mg/100gfw) | Carotenoids (mg/gfw) | Total Chl. (mg/gfw) | Chl. b (mg/gfw) | Chl. a (mg/gfw) | Sugars (mg/gfw) | Proline (mg/gfw) | Protein (mg/gfw) |
| Province | 6 | 64.820** | 3962.076** | 4.306** | 15.436** | 1.387** | 21.086** | 2.722** | 5.66E-03** | 0.033 |
| Populations (Province) | 8 | 11.623** | 690.901** | 3.038** | 10.766** | 0.428** | 4.295** | 0.548** | 4.79E-03** | 0.006 |
| Error | 135 | 0.367 | 0.873 | 0.002 | 0.009 | 0.002 | 0.003 | 0.003 | 4.28E-05 | 2.3E-07 |
| CV of design (%) | | 8.83 | 1.47 | 0.99 | 0.92 | 1.73 | 0.68 | 1 | 0.98 | 0.18 |

**: Significant at 1% probability level.

to the values of the PCV of the traits, it is observed that there is a favorable variation among the traits for most of them, and they can be used to improve the studied populations of sickleweed (Table 2). The lowest PCV related to protein traits (0.23), and the highest related to phenol (16.45) and other traits were in this range. The reason for the high diversity of traits can be the environmental conditions and the different genetic backgrounds of the populations. The variation in leaf length trait and petiole length trait was more than other traits and therefore these traits can be considered by breeders because these traits play a role in stress tolerance.

The results of variance analysis of the unbalanced nest design for physiological and biochemical traits as well as the percentage of essential oil are shown in Table 3. The effects of the province and the population within the province for the studied traits showed a significant difference at the level of 0.01 between sickleweed populations (Table 3). The significance of the effects of the province and the population within the province showed that there is a significant difference between the provinces and between 15 different populations, and it shows the diversity among these populations and provinces.

Table 4 shows the results of variance components for the studied traits and the results showed that the percentage of variance components for the studied traits was between 25.31 and 83.92% among the studied populations and between 8.51 and 73.36% among the provinces. Based on these results, considerable diversity is observed between populations and between provinces and these differences can be used to select the best population. In addition, for some traits, variation within the population was also observed, which sees from the results of the percentage of variance components for the source of error. Variation within the population for these traits was less than 8%. Therefore, the chance of selecting superior individuals within populations will be low due to low diversity.

**Table 4. Estimation of the variance components of the sources of variation in the unbalanced nest design of the studied physiological and biochemical traits.**

| Traits | Sources of variation | | | | | |
|---|---|---|---|---|---|---|
| | Province | | Populations (Province) | | Error | |
| | Variance Components | % | Variance Components | % | Variance Components | % |
| Chl. a (mg/gfw) | 0.933 | 68.354 | 0.429 | 31.451 | 0.003 | 0.196 |
| Chl. b (mg/gfw) | 0.053 | 54.538 | 0.043 | 43.639 | 0.002 | 1.823 |
| Total Chl. (mg/gfw) | 0.259 | 19.307 | 1.076 | 80.054 | 0.009 | 0.639 |
| Carotenoids (mg/gfw) | 0.070 | 18.742 | 0.304 | 80.816 | 0.002 | 0.442 |
| Proline (mg/gfw) | 0.0015 | 73.356 | 0.0006 | 26.633 | 0.00000023 | 0.011 |
| Protein (mg/gfw) | 0.000048 | 8.506 | 0.000475 | 83.928 | 0.0000428 | 7.567 |
| Sugars (mg/gfw) | 0.121 | 67.838 | 0.054 | 30.601 | 0.003 | 1.561 |
| Phenol (mg/100gfw) | 181.732 | 72.228 | 69.003 | 27.425 | 0.873 | 0.347 |
| Essential oil (%) | 2.955 | 66.45 | 1.126 | 25.31 | 0.367 | 8.24 |

**Table 5. The mean comparison of physiological and biochemical studied traits for sickleweed populations using Duncan's method.**

| Populations | Traits | | | | | | | | |
|---|---|---|---|---|---|---|---|---|---|
| | Essential oil (%) | Phenol | Sugars (mg/ gfw) | Protein (mg/ gfw) | Proline (mg/ gfw) | Carotenoids (mg/ gfw) | Total Chl. (mg/ gfw) | Chl. b (mg/ gfw) | Chl. a (mg/ gfw) |
| Kivananat | 8.90ab | 53.545h | 5.656c | 0.684bc | 0.261i | 4.152f | 9.597h | 2.216h | 7.381g |
| Bavaleh | 7.69cd | 53.411h | 5.712bc | 0.683bc | 0.271f | 4.597d | 10.892d | 2.325fg | 8.377d |
| Sahneh | 8.55a-c | 49.793i | 5.398d | 0.6706d | 0.287e | 4.778b | 8.173k | 2.074i | 7.228h |
| Chaharduli | 7.32de | 44.995j | 5.006f | 0.697a | 0.245k | 3.417j | 9.602h | 2.401e | 6.099m |
| Panjeh Ali | 5.43f | 66.269f | 5.016f | 0.638f | 0.227mn | 4.848a | 10.516e | 2.261gh | 8.255e |
| Bolbanabad | 4.36gh | 69.503e | 4.836g | 0.621g | 0.249j | 3.014l | 9.778g | 2.027i | 7.752f |
| Amirabad | 6.52e | 86.645b | 5.013f | 0.659e | 0.266h | 3.963g | 9.094j | 2.218h | 6.876j |
| Gerger-e Sofla | 7.11de | 81.538c | 5.272e | 0.676cd | 0.228m | 3.650i | 11.714b | 2.329f | 7.116i |
| Qaleh Gah | 5.26fg | 88.149a | 5.424d | 0.693ab | 0.269g | 4.167f | 9.182j | 2.492d | 6.224l |
| nemat abad auliya | 4.18h | 65.539f | 4.699h | 0.656e | 0.294d | 3.845h | 10.009f | 2.717c | 7.293h |
| Seylatan | 4.03h | 73.589d | 5.201e | 0.679cd | 0.227n | 3.231k | 9.588h | 2.516d | 6.791k |
| Ilan | 9.25a | 61.788g | 4.817g | 0.659e | 0.237l | 4.697c | 9.350i | 2.797b | 6.858kj |
| Shaban | 9.35a | 43.489k | 5.949a | 0.676cd | 0.366a | 4.907a | 9.441i | 2.958a | 9.384b |
| Khalajastan | 6.59e | 50.555i | 5.768b | 0.635f | 0.310c | 4.317e | 11.282c | 2.415e | 8.867c |
| Deylaman | 8.28bc | 66.685f | 5.033f | 0.642f | 0.342b | 4.141f | 12.482a | 2.919a | 9.562a |

The treatments that have the same letters do not differ statistically at the 1% level.

Based on the mean comparison results, different sickleweed populations were placed in different groups, which indicates the diversity and increased chance of selection for breeders. The results of the mean comparison for the studied traits showed that there is a big difference between different populations of sickleweed and this situation increases the chance of choosing favorable sickleweed populations for the breeder. The Ard-Shaban population had the highest values for essential oil, phenol, total chlorophyll, and Chl. a, and Chl. traits, and follow in the Qaz-Ilan population (Table 5). Therefore, these populations can be recommended for planting in different areas due to the high performance of photosynthesis, which plays a role in increasing production.

The results of correltion analysis are presented in Fig 2. Based on the results, there was a significant positive correlation between essential oils and sugars, proline, carotenoids and chlorophylls. There was also a negative correlation between essential oils and phenols. There was a positive correlation between sugars and protein, proline and carotenoids and Chl. a.

Principal component analysis was used before cluster analysis to clarify the relative importance of the variables that contribute to the cluster. The results of PCA for the studied traits are shown in Table 6. The eigenvalues obtained from components 1 to 3 justified 97.6, 1.2, and 0.85% of the total variance, respectively. The coefficients of the eigenvectors obtained from the first component showed that the phenol trait was the most important trait for clustering in this component. In the second component, Essential oil and Carotenoids traits, and in the third component, Total Chl. and Chl. a traits were the most important traits. The distribution of the studied populations based on the first two components is shown in Fig 3. The studied populations were divided into three groups. Populations Kur-Gerger-e, Sofla, Kur-Amirabad, and Kur-Qaleh Gah were in the first group, populations Kur-Panjeh Ali, Kur-Seylatan, Kur-Nemat abad auliya, Kur-Bolbanabad, Ham-Chaharduli, and Qom-Khalajastan were in the second group, and the rest of the populations were in the third group. Since the role of phenol in the first component and the role of essential oil and carotenoids in the second component was

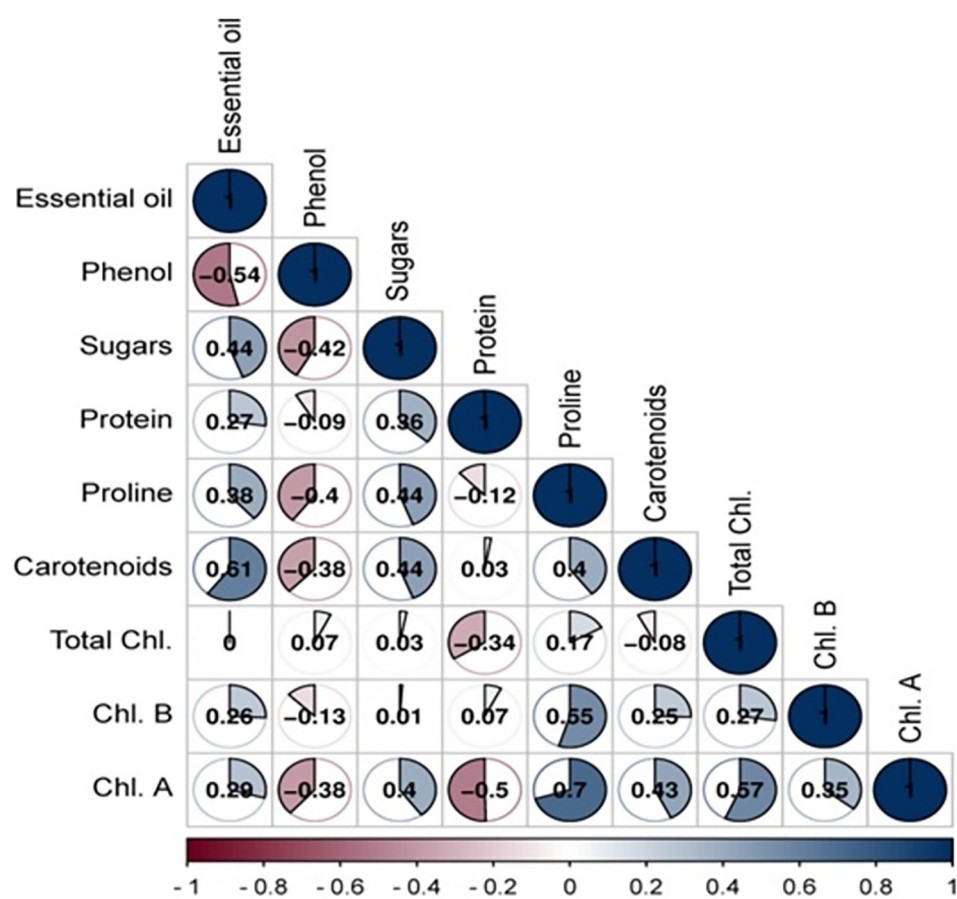

**Fig 2. The correlation diagram of studied traits in sickleweed populations.**

positive and effective. Therefore, the populations placed in the first area of the biplot are suitable populations for increasing the percentage of essential oil as well as carotenoids and phenol. In addition, biochemical and physiological traits play a role in stress tolerance, therefore the first area of the biplot of PC1 and PC3 is suitable. The Gilan-Deylaman and Kur-Gerger-e

**Table 6. The results of four principal component analyses for traits.**

| Traits | PC1 | PC2 | PC3 | PC4 |
|---|---|---|---|---|
| Essential oil | -0.06961 | 0.93826 | -0.25872 | -0.15457 |
| Phenol | 0.99697 | 0.07436 | -0.00513 | 0.019862 |
| Sugars | -0.01111 | 0.072748 | 0.022018 | 0.15669 |
| Protein | -0.00014 | 0.001947 | -0.00908 | -0.01002 |
| Proline | -0.00115 | 0.007903 | 0.010366 | 0.026353 |
| Carotenoids | -0.01557 | 0.19925 | -0.01841 | 0.46504 |
| Total Chl. | 0.005571 | 0.16237 | 0.75521 | -0.56126 |
| Chl. b | -0.0026 | 0.053882 | 0.056897 | 0.025628 |
| Chl. a | -0.02838 | 0.19951 | 0.5987 | 0.64686 |
| Eigenvalue | 210.545 | 2.59209 | 1.82469 | 0.474473 |
| % variance | 97.593 | 1.2015 | 0.84579 | 0.21993 |

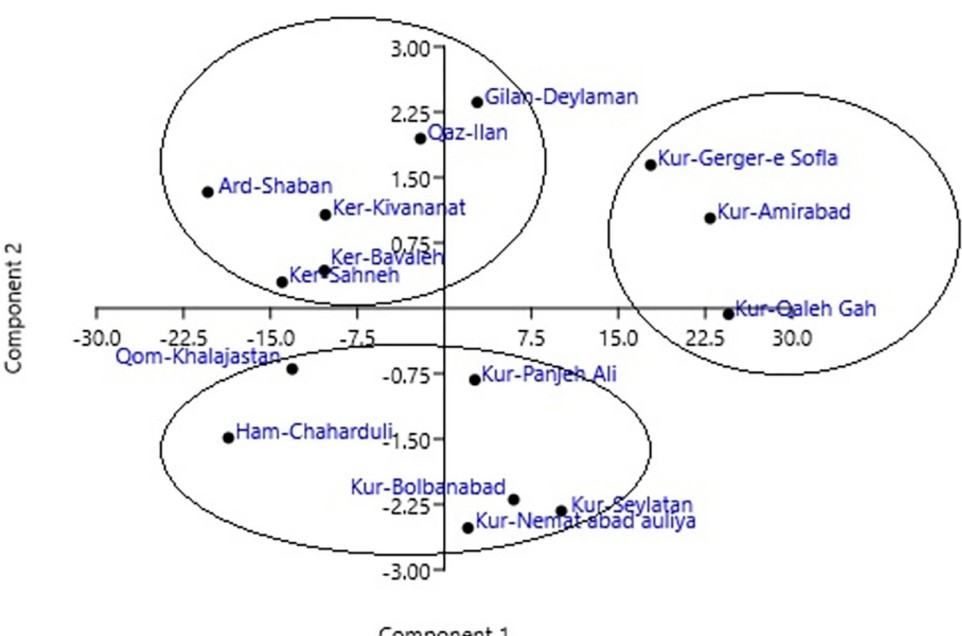

**Fig 3. The biplot grouping of studied population based on four principal component analysis.**

Sofla populations were placed in this area and identified as the superior populations of this study and can be used in breeding programs for tolerance.

To determine the degree of diversity among the studied sickleweed populations was used the cluster analysis method. Different cluster analysis methods with different distance criteria were used to determine the degree of diversity between the studied populations, and the results showed that the UPGMA method with Euclidean distance had the highest value of the Cophentic correlation coefficient (0.79), and therefore, the cluster analysis was performed with this method.

The results of cluster analysis showed that there was the lowest genetic distance (2.08) between the populations of Ker-Bavaleh and Ker-Kivananat and the highest genetic distance (44.97) was between the populations of Ard-Shaban and Kur-Qaleh Gah. As seen in Fig 4, the different populations of sickleweed are divided into three main groups. The first group included the population of Kur-Gerger-e, Sofla, Kur-Qaleh, Gah, and Kur-Amirabad, the second group included the population of Kur-Panjeh Ali, Kur Seylatan, Kur-Nemat abad auliya, Kur-Bolbanabad, Gilan-Deylaman, and Qaz-Ilan, and In the third group were the populations of Ker-Kivananat, Ker-Bavaleh, Ker-Sahneh, Ard-Shaban, Ham-Chaharduli, and Qom-Khalajastan. The results showed that the populations were placed in separate categories due to different genetic bases or other environmental factors, and it can be justified that the morphological traits were able to determine this distinction. The results of cluster analysis show the differences between the populations of each group and the similarities and kinships of the populations within each group. The reason for the difference between the populations of the groups can be due to the difference in the genetic structure or the effect of other environmental factors on the traits.

Population grouping by PCA was highly consistent with the grouping of cluster analysis. The results of this experiment showed that the number of studied traits is different in different regions. This diversity and difference in the amount of biochemical and physiological traits of sickleweed plants in different regions of Iran and other parts of the world can be caused by

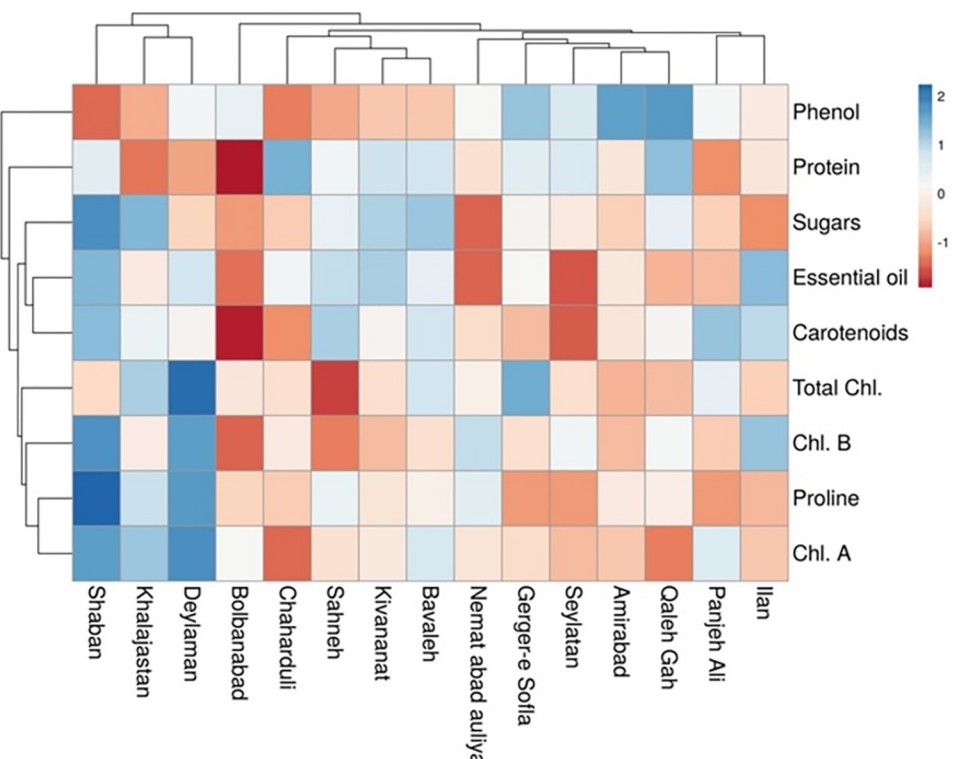

**Fig 4. The cluster analysis dendrogram of studied population based on UPGMA method.**

different weather factors (light, temperature, precipitation, relative humidity, etc.), geographical factors (geographical height, longitude, and latitude), soil conditions (soil texture, materials and nutrients) and genetic factors. Climatic conditions such as light, altitude, and average temperature have a great effect on the production of chemical compounds in horticultural and medicinal products [47]. The amount and type of active substances in medicinal plants are determined by the guidance of both environmental and genetic factors [48].

The investigation of phytochemical and physiological traits in this research showed that there was a significant difference between the studied traits of phenol content, percentage of the essence, biochemical, and physiological traits in different regions of Iran, in such a way that populations Gilan-Deylaman and Kur-Gerger-e Sofla have the highest values for these evaluated traits. Separate studies have shown in different medicinal plants [17,18,20] that biochemical and physiological trait are influenced by different collection areas, which is consistent with the results of this study. In the study of Crataegus oxyacantha plants, it was shown that the place of growth, height, and type of organ has a significant effect on the amount of phenol and flavonoids and it was found that at higher altitudes, more phenol and flavonoid compounds are produced in the plant [49–51]. Of course, the difference in the quantitative amounts of phytochemical compounds, including phenolic and flavonoid compounds, among the populations of different regions can be caused by genetic diversity or ecological conditions governing the habitats [51,52]. The choosing and developing plants with high medicinal potential, high yield, or resilience to environmental stress requires an understanding of the variance in biochemical and physiological properties of plants across different geographic and climatic locations. Understanding the ecological and evolutionary processes that influence plant diversity and environment adaption is also crucial [24,25].

## Conclusion

Since the high level of proline and biochemical and physiological traits in plants can play a role in plants' tolerance to stresses, therefore, populations with high values of these traits can be used in stress tolerance breeding programs. In addition, the essential oil of this plant is used in the treatment of diseases, and therefore populations Ard-Shaban and Qaz-Ilan, which showed a high percentage of essential oil and Gilan-Deylaman and Kur-Gerger-e Sofla were identified as superior populations that they can be used in breeding programs. In conclusion, due to the influence of climatic conditions including temperature, rainfall, soil type, and altitude, the biochemical and physiological characteristics of medicinal plants might differ dramatically across different areas. The therapeutic potential of medicinal plants is significantly impacted by the diversity in their biochemical and physiological characteristics. As a result, it's critical to comprehend how diverse geographical areas affect medicinal plants and to build plans for their production and application in the pharmaceutical sector.

## Author Contributions

**Formal analysis:** Mehdi Rahimi.

**Investigation:** Mehdi Rahimi, Mojtaba Kordrostami, Jaber Nasiri.

**Writing – original draft:** Mehdi Rahimi, Mojtaba Kordrostami, Jaber Nasiri.

**Writing – review & editing:** Mehdi Rahimi, Mojtaba Kordrostami, Jaber Nasiri.

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
