## [Decision Letter · Decision Letter 0]

1 Mar 2023

PONE-D-23-02750Evaluation of biochemical, physiological traits and percentage of essential oil of sickleweed (Falcaria vulgaris) population in different geographical and climatic regionsPLOS ONE

Dear Dr. Rahimi,

Thank you for submitting your manuscript to PLOS ONE. After careful consideration, we feel that it has merit but does not fully meet PLOS ONE’s publication criteria as it currently stands. Therefore, we invite you to submit a revised version of the manuscript that addresses the points raised during the review process.

ACADEMIC EDITOR:

Abstract of the manuscript must be revised highlighting the methodology and results clearly. This section should contain all key results, and conclusion which is missing.

Introduction needs some careful attention. All the abbreviations on first appearance must be presented in full. Language is generally poor. In some places is it not quite clear what the authors meant to say. Line 13 perennial and perennial??? Line 14 genus Falcaria. Genus should be italicized. Moreover, population Gilan-Deylaman and Kur- 29 Gerger-e Sofla were also identified as superior populations??? Not clear. Units are missing in different parts of manuscript like in table 1 Height above sea level. Author mentioned in line 64 the importance of Identification of morphological diversity but not describe in the manuscript. Climatic factors are not investigated as a main objective of your work.   

Material and methods need improvement. This section is weak, we need complete procedures. All phytochemical analyzed in the study should be explained separately with subheadings like: Sample collection technique (plant collection): Sample Preparation: as each assay has a specific method of sample preparation or extraction. Phytochemicals: Each assay should be explained in this section with subheadings such as, Total chlorophyll: Total carotenoids:

Results are not discussed in appropriately. Discussion needs improvement with more references to support your findings.

Conclusions are not in support of your findings. Significant findings should be mentioned with recommendation.

Please submit your revised manuscript by Apr 15 2023 11:59PM. If you will need more time than this to complete your revisions, please reply to this message or contact the journal office at plosone@plos.org. Please include the following items when submitting your revised manuscript:

We look forward to receiving your revised manuscript.

Kind regards,

Alia Ahmed

Academic Editor

PLOS ONE

“We gratefully acknowledge the research funding provided for this project (No. 99026928) by Iran National Science Foundation (INSF).”

Reviewers' comments:

Reviewer's Responses to Questions

**Comments to the Author**

1. Is the manuscript technically sound, and do the data support the conclusions?

Reviewer #1: Partly

Reviewer #2: Yes

2. Has the statistical analysis been performed appropriately and rigorously? 

Reviewer #1: Yes

Reviewer #2: Yes

3. Have the authors made all data underlying the findings in their manuscript fully available?

Reviewer #1: Yes

Reviewer #2: Yes

4. Is the manuscript presented in an intelligible fashion and written in standard English?

Reviewer #1: Yes

Reviewer #2: Yes

5. Review Comments to the Author

Reviewer #1: The authors conducted a study entitled "Evaluation of biochemical, physiological traits and percentage of essential oil of sickleweed (Falcaria vulgaris) population in different geographical and climatic regions"

I regret to inform you that I cannot recommend this work for publication in this prestigious journal.

For me the study is limited and important information should be inserted in the MS, the chemical composition of the essential oils should be investigated, a practical application of these oils and the correlation with the different collection times should be investigated, the information brought in the work is not sufficient to justify its publication.

Reviewer #2: Although samples from different environments cannot fully reflect the true genetic information, it also can screen out the most suitable populations of the same species in a given place.

There are still some minor problems in the manuscript that need further revision

1.In the introduction, the goal of breeding is not stated clearly. In line 84, what aspects of the best populations with breeding potential are considered and what are the criteria? It should be stated clearly in the preceding introduction.

2. in L151, what is the full name of PCV? phenotypic coefficient variation？Abbreviations should be given where the manuscript first appears

3.in L213-226, the specific traits of the three main groups are not described separately and are not marked in Figure 1.

4.The results in this manuscript are not well compared and analyzed with those already reported.

5.There is no detailed discussion on how the suitable or superior population is judged and compared, and what are their distinguishing features from other populations.

6. PLOS authors have the option to publish the peer review history of their article (what does this mean?). If published, this will include your full peer review and any attached files.

Reviewer #1: No

Reviewer #2: No

---

## [Author Response · Author response to Decision Letter 0]

21 Mar 2023

Dear Editor

Thank you for your useful comments and suggestions on the language and structure of our manuscript. We have modified the manuscript accordingly and showed in the revised MS by yellow highlighted and Track Changes. Also, detailed corrections are listed below point by point.

Thanks once again.

Best Regards

On behalf of the authors

ACADEMIC EDITOR:

Abstract of the manuscript must be revised highlighting the methodology and results clearly. This section should contain all key results, and conclusion which is missing.

Introduction needs some careful attention. All the abbreviations on first appearance must be presented in full. Language is generally poor. In some places is it not quite clear what the authors meant to say. Line 13 perennial and perennial??? Line 14 genus Falcaria. Genus should be italicized. Moreover, population Gilan-Deylaman and Kur- 29 Gerger-e Sofla were also identified as superior populations??? Not clear. Units are missing in different parts of manuscript like in table 1 Height above sea level. Author mentioned in line 64 the importance of Identification of morphological diversity but not describe in the manuscript. Climatic factors are not investigated as a main objective of your work. 

Material and methods need improvement. This section is weak, we need complete procedures. All phytochemical analyzed in the study should be explained separately with subheadings like: Sample collection technique (plant collection): Sample Preparation: as each assay has a specific method of sample preparation or extraction. Phytochemicals: Each assay should be explained in this section with subheadings such as, Total chlorophyll: Total carotenoids:

Results are not discussed in appropriately. Discussion needs improvement with more references to support your findings.

Conclusions are not in support of your findings. Significant findings should be mentioned with recommendation.

Answer: All the things requested by the respected editor were done in the text and the article was corrected accordingly.

Reviewer comments

Reviewer #1

The authors conducted a study entitled "Evaluation of biochemical, physiological traits and percentage of essential oil of sickleweed (Falcaria vulgaris) population in different geographical and climatic regions"

I regret to inform you that I cannot recommend this work for publication in this prestigious journal.

 For me the study is limited and important information should be inserted in the MS, the chemical composition of the essential oils should be investigated, a practical application of these oils and the correlation with the different collection times should be investigated, the information brought in the work is not sufficient to justify its publication.

Answer: In this article, the aim was not to investigate the chemical composition of the essential oil, and of course the opinion of the respected referee is correct and it can be a new article. The main goal was to investigate the effect of environments with different altitude, temperature and climate on biochemical traits and percentage of essential oil.

Graduate University of Advanced Technology (Kerman, Iran) to support this project.

Reviewer #2

Reviewer #2: Although samples from different environments cannot fully reflect the true genetic information, it also can screen out the most suitable populations of the same species in a given place. There are still some minor problems in the manuscript that need further revision

1.In the introduction, the goal of breeding is not stated clearly. In line 84, what aspects of the best populations with breeding potential are considered and what are the criteria? It should be stated clearly in the preceding introduction.

Answer: Corrected and revised in text

2. in L151, what is the full name of PCV? phenotypic coefficient variation？Abbreviations should be given where the manuscript first appears

Answer: Corrected in text

3.in L213-226, the specific traits of the three main groups are not described separately and are not marked in Figure 1.

Answer: Corrected in text

4.The results in this manuscript are not well compared and analyzed with those already reported.

Answer: Corrected in text

5.There is no detailed discussion on how the suitable or superior population is judged and compared, and what are their distinguishing features from other populations.

Answer: revised in text

We would like to give you a special thanks for your careful reading of the whole MS and also for your meaningful corrections that helped us a lot to improve our manuscript significantly. We have revised the manuscript according to your suggestions and the other reviewer’s suggestions and highlighted.

Best Regards

On behalf of the authors

---

## [Decision Letter · Decision Letter 1]

12 May 2023

PONE-D-23-02750R1Evaluation of biochemical, physiological traits and percentage of essential oil of sickleweed (Falcaria vulgaris) population in different geographical and climatic regionsPLOS ONE

Dear Dr. Rahimi,

Thank you for submitting your manuscript to PLOS ONE. After careful consideration, we feel that it has merit but does not fully meet PLOS ONE’s publication criteria as it currently stands. Therefore, we invite you to submit a revised version of the manuscript that addresses the points raised during the review process.

ACADEMIC EDITOR: Please ensure that your decision is justified on PLOS ONE’s publication criteria and not, for example, on novelty or perceived impact.

We look forward to receiving your revised manuscript.

Kind regards,

Alia Ahmed

Academic Editor

PLOS ONE

Journal Requirements:

Reviewers' comments:

Reviewer's Responses to Questions

**Comments to the Author**

1. If the authors have adequately addressed your comments raised in a previous round of review and you feel that this manuscript is now acceptable for publication, you may indicate that here to bypass the “Comments to the Author” section, enter your conflict of interest statement in the “Confidential to Editor” section, and submit your "Accept" recommendation.

Reviewer #3: All comments have been addressed

Reviewer #4: (No Response)

2. Is the manuscript technically sound, and do the data support the conclusions?

Reviewer #3: Yes

Reviewer #4: Yes

3. Has the statistical analysis been performed appropriately and rigorously? 

Reviewer #3: Yes

Reviewer #4: Yes

4. Have the authors made all data underlying the findings in their manuscript fully available?

Reviewer #3: Yes

Reviewer #4: Yes

5. Is the manuscript presented in an intelligible fashion and written in standard English?

Reviewer #3: Yes

Reviewer #4: Yes

6. Review Comments to the Author

Reviewer #3: Authors have done nice work on the 15 different sickleweed populations from seven provinces of the Iran. The data obtained will be helpful to choose sickleweed population in breeding programs. The work showed that there is variation among populations which is very important for further planning of research on sickleweed.

I recommend the paper for publication after following corrections.

Do following corrections:

Line Nos.:

139, hour to h

142, minute to min

146-147, degree Celsius to OC

154, two milliliters to 2 ml

156, degrees Celsius to OC

157, minute to min

157, two milliliters to 2 ml

199, started line with 19 bioclimatic indicators instead start with, “Bioclimatic indicators (19)”.

Conclusion is very lengthy and bit vague. Make is concise and to the point.

Reviewer #4: The authors should address the following:

• What are the ages of different Sickleweed plant obtained from each region?

• What is the reference of formula 1, 2, 3 and 4?

• Give some brief information for Elbow technique which used in clustering the regions?

• Conclusion should be shortened

• See the attached MS file and consider all required corrections

7. PLOS authors have the option to publish the peer review history of their article (what does this mean?). If published, this will include your full peer review and any attached files.

Reviewer #3: No

Reviewer #4: **Yes: **Hani Saudy

---

## [Author Response · Author response to Decision Letter 1]

30 May 2023

Dear Editor

Thank you for your useful comments and suggestions on the language and structure of our manuscript. We have modified the manuscript accordingly and showed in the revised MS by yellow highlighted and Track Changes. Also, detailed corrections are listed below point by point.

Thanks once again.

Best Regards

On behalf of the authors

Reviewer comments

Reviewer #3:

Authors have done nice work on the 15 different sickleweed populations from seven provinces of the Iran. The data obtained will be helpful to choose sickleweed population in breeding programs. The work showed that there is variation among populations which is very important for further planning of research on sickleweed.

I recommend the paper for publication after following corrections.

Do following corrections:

Line Nos.:139, hour to h

Answer: Corrected in text

142, minute to min

Answer: Corrected in text

146-147, degree Celsius to OC

Answer: Corrected in text

154, two milliliters to 2 ml

Answer: Corrected in text

156, degrees Celsius to OC

Answer: Corrected in text

157, minute to min

Answer: Corrected in text

157, two milliliters to 2 ml

Answer: Corrected in text

199, started line with 19 bioclimatic indicators instead start with, “Bioclimatic indicators (19)”.

Answer: Corrected in text

Conclusion is very lengthy and bit vague. Make is concise and to the point.

Answer: The conclusion was summarized.

Reviewer #4

The authors should address the following:

What are the ages of different Sickleweed plant obtained from each region?

Answer: Corrected and revised in text. Although Sickleweed has been described as annual, biennial or perennial [13], but according to the authors' observations and evaluations, it was annual.

What is the reference of formula 1, 2, 3 and 4?

Answer: Corrected in text and the reference added to text.

Give some brief information for Elbow technique which used in clustering the regions?

Answer: Added in text and describe this method.

Conclusion should be shortened

Answer: The conclusion was summarized.

See the attached MS file and consider all required corrections

Answer: All required corrections in text based on suggestion and Also, suggested references were added to the text.

We would like to give you a special thanks for your careful reading of the whole MS and also for your meaningful corrections that helped us a lot to improve our manuscript significantly. We have revised the manuscript according to your suggestions and the other reviewer’s suggestions and highlighted.

Best Regards

On behalf of the authors

---

## [Decision Letter · Decision Letter 2]

6 Jun 2023

Evaluation of biochemical, physiological traits and percentage of essential oil of sickleweed (Falcaria vulgaris) population in different geographical and climatic regions

PONE-D-23-02750R2

Dear Dr. Rahimi,

We’re pleased to inform you that your manuscript has been judged scientifically suitable for publication and will be formally accepted for publication once it meets all outstanding technical requirements.

Kind regards,

Alia Ahmed

Academic Editor

PLOS ONE

Additional Editor Comments (optional):

Reviewers' comments:

Reviewer's Responses to Questions

**Comments to the Author**

1. If the authors have adequately addressed your comments raised in a previous round of review and you feel that this manuscript is now acceptable for publication, you may indicate that here to bypass the “Comments to the Author” section, enter your conflict of interest statement in the “Confidential to Editor” section, and submit your "Accept" recommendation.

Reviewer #4: All comments have been addressed

2. Is the manuscript technically sound, and do the data support the conclusions?

Reviewer #4: Yes

3. Has the statistical analysis been performed appropriately and rigorously? 

Reviewer #4: Yes

4. Have the authors made all data underlying the findings in their manuscript fully available?

Reviewer #4: Yes

5. Is the manuscript presented in an intelligible fashion and written in standard English?

Reviewer #4: Yes

6. Review Comments to the Author

Reviewer #4: frankly, the authors performed the correction in good manner. all issues have been addressed

the manuscript technically is sound, and do the data support the conclusions

the statistical analysis has been performed appropriately and rigorously

7. PLOS authors have the option to publish the peer review history of their article (what does this mean?). If published, this will include your full peer review and any attached files.

Reviewer #4: No

---

## [Editor Report · Acceptance letter]

12 Jun 2023

PONE-D-23-02750R2 

Evaluation of biochemical, physiological traits and percentage of essential oil of sickleweed (*Falcaria vulgaris*) population in different geographical and climatic regions 

Dear Dr. Rahimi:

I'm pleased to inform you that your manuscript has been deemed suitable for publication in PLOS ONE. Congratulations! Your manuscript is now with our production department. 

Kind regards, 

on behalf of

Dr. Alia Ahmed 

Academic Editor

PLOS ONE